# Growth Response and Differentiation of Bone Marrow-Derived Mesenchymal Stem/Stromal Cells in the Presence of Novel Multiple Myeloma Drug Melflufen

**DOI:** 10.3390/cells11091574

**Published:** 2022-05-07

**Authors:** Arjen Gebraad, Roope Ohlsbom, Juho J. Miettinen, Promise Emeh, Toni-Karri Pakarinen, Mikko Manninen, Antti Eskelinen, Kirsi Kuismanen, Ana Slipicevic, Fredrik Lehmann, Nina N. Nupponen, Caroline A. Heckman, Susanna Miettinen

**Affiliations:** 1Adult Stem Cell Group, Faculty of Medicine and Health Technology, Tampere University, 33520 Tampere, Finland; roope.ohlsbom@tuni.fi (R.O.); promise.emeh@tuni.fi (P.E.); susanna.miettinen@tuni.fi (S.M.); 2Research, Development and Innovation Centre, Tampere University Hospital, 33520 Tampere, Finland; 3Institute for Molecular Medicine Finland-FIMM, HiLIFE–Helsinki Institute of Life Science, iCAN Digital Precision Cancer Medicine Flagship, University of Helsinki, 00290 Helsinki, Finland; juho.miettinen@helsinki.fi (J.J.M.); caroline.heckman@helsinki.fi (C.A.H.); 4Department of Musculoskeletal Diseases, Tampere University Hospital, 33520 Tampere, Finland; toni-karri.pakarinen@pshp.fi; 5Orton Orthopaedic Hospital, 00280 Helsinki, Finland; mikko.manninen@orton.fi; 6Coxa Hospital for Joint Replacement, Faculty of Medicine and Health Technology, Tampere University, 33520 Tampere, Finland; antti.eskelinen@coxa.fi; 7Department of Obstetrics and Gynecology, Tampere University Hospital, 33520 Tampere, Finland; kirsi.kuismanen@pshp.fi; 8Oncopeptides AB, 111 37 Stockholm, Sweden; slipicevic@yahoo.no (A.S.); fredrik.lehmann@oncopeptides.com (F.L.); nnnupponen@gmail.com (N.N.N.)

**Keywords:** melflufen, melphalan flufenamide, peptide–drug conjugate, mesenchymal stem/stromal cells, multiple myeloma, bone marrow, drug sensitivity, osteogenesis, adipogenesis, angiogenesis

## Abstract

Mesenchymal stem/stromal cells (MSCs) are self-renewing and multipotent progenitors, which constitute the main cellular compartment of the bone marrow stroma. Because MSCs have an important role in the pathogenesis of multiple myeloma, it is essential to know if novel drugs target MSCs. Melflufen is a novel anticancer peptide–drug conjugate compound for patients with relapsed refractory multiple myeloma. Here, we studied the cytotoxicity of melflufen, melphalan and doxorubicin in healthy human bone marrow-derived MSCs (BMSCs) and how these drugs affect BMSC proliferation. We established co-cultures of BMSCs with MM.1S myeloma cells to see if BMSCs increase or decrease the cytotoxicity of melflufen, melphalan, bortezomib and doxorubicin. We evaluated how the drugs affect BMSC differentiation into adipocytes and osteoblasts and the BMSC-supported formation of vascular networks. Our results showed that BMSCs were more sensitive to melflufen than to melphalan. The cytotoxicity of melflufen in myeloma cells was not affected by the co-culture with BMSCs, as was the case for melphalan, bortezomib and doxorubicin. Adipogenesis, osteogenesis and BMSC-mediated angiogenesis were all affected by melflufen. Melphalan and doxorubicin affected BMSC differentiation in similar ways. The effects on adipogenesis and osteogenesis were not solely because of effects on proliferation, seen from the differential expression of differentiation markers normalized by cell number. Overall, our results indicate that melflufen has a significant impact on BMSCs, which could possibly affect therapy outcome.

## 1. Introduction

Multiple myeloma is a B-cell lineage cancer in which neoplastic plasma cells expand in the bone marrow. Although the current treatments are highly effective in controlling the disease and producing deep remissions, the disease invariably relapses after a period of time, requiring continued therapeutic intervention to maintain disease control [1].

Mesenchymal stem/stromal cells (MSCs) are self-renewing and multipotent progenitors that can differentiate into a variety of cell types, such as adipocytes, osteoblasts and pericytes, which constitute the main cellular compartment of bone marrow stroma [2,3].

Many studies have demonstrated that MSCs support the proliferation, survival, migration and drug resistance of myeloma cells [4,5,6]. MSCs are also involved in angiogenesis as vasculature-supporting pericytes, thereby supporting the tumor’s blood supply [7]. Moreover, the activity of MSC-derived osteoblasts decreases in multiple myeloma, causing osteolytic lesions and fractures [8].

Knowledge on the interactions between myeloma cells and MSCs, adipocytes, pericytes and osteoblasts is crucial to understanding how tumors grow within the bone marrow and how osteolytic lesions form. Because of the important role of MSCs in the pathogenesis of multiple myeloma, it is essential to know if novel drugs target MSCs.

Alkylating agents represent the oldest class of anticancer drugs, acting through covalent interaction with cellular macromolecules such as DNA. Melphalan is a classic alkylator and has been in clinical use for approximately 50 years in a wide variety of malignancies, including multiple myeloma [9].

Melphalan flufenamide, hereafter called melflufen, is an alkylating prodrug of melphalan. Despite identical alkylating capacity, melflufen exhibits significantly higher in vitro and in vivo cytotoxicity than melphalan itself [10,11,12,13,14,15]. Melflufen diffuses rapidly through the cell membranes because of its high lipophilicity. Once inside the cell, melflufen is rapidly hydrolyzed by peptidases and esterases and metabolized to active alkylating molecules [16,17]. Because of its novel mechanism of action, melflufen has shown clinically meaningful benefits to patients with relapsed/refractory multiple myeloma [18].

Here, we studied the cytotoxicity of melflufen in healthy human bone marrow-derived MSCs (BMSCs) and how melflufen affects BMSC proliferation. We established co-cultures of BMSCs with MM.1S myeloma cells to see if they increase or decrease the tumoricidal effect of melflufen. We also evaluated how melflufen affects the differentiation of BMSCs into adipocytes and osteoblasts and BMSC-supported formation of vascular networks. Melphalan and doxorubicin were used as control drugs. Bortezomib was included as the control drug in co-cultures with MM.1S myeloma cells.

## 2. Materials and Methods

### 2.1. Bone Marrow-Derived Mesenchymal Stem Cell Isolation and Culture

BMSCs were isolated from human bone marrow aspirate samples obtained from 8 donors (Appendix A) during surgical procedures at the Department of Orthopedics and Traumatology at Tampere University Hospital, Orton Orthopaedic Hospital and Coxa Hospital for Joint Replacement, with the patient’s consent and as described previously with slight modifications [19,20]. Bone marrow was obtained under approval of the Tampere University Hospital Ethics Committee, Tampere, Finland (R15174). The bone marrow was diluted 1:3 with Dulbecco’s phosphate buffered saline (DPBS, Gibco™, Thermo Fisher Scientific, Waltham, MA, USA). The mixture was layered on a Histo Paque-1077 (Sigma-Aldrich, Manassas, VA, USA) cushion and centrifuged at 800× *g* for 20 min at room temperature. Mononuclear cells were collected from the liquid interface and washed with Minimum Essential Medium α (α MEM, Gibco™). BMSCs were obtained as the adherent fraction after expansion in basic medium consisting of α MEM, 5% human serum (HS; BioWest, Nuaillé, France), and 1% antibiotics (100 U/mL penicillin; 100 U/mL streptomycin; Lonza, Basel, Switzerland) with 5 ng/mL of human FGF-2 (Miltenyi Biotec; Bergisch Gladbach, Germany) at 37 °C and 5% CO_2_ in a humidified atmosphere. Medium was changed twice per week. Cells were detached at 75% confluency with TrypLE Select (Thermo Fisher Scientific) and either passaged or cryopreserved in vapor phase nitrogen.

To verify the mesenchymal origin of the isolated BMSCs, surface marker expression was characterized by flow cytometry (FACSAria Fushion; BD Biosciences, Erembodegem, Belgium) as described previously [21] (Appendix A). A large majority of cells (>85%) expressed CD73 (ecto-50-nucleotidase), CD90 (Thy-1), CD105 (endoglin), and HLA-DR (leukocyte antigen). Only a small portion (<10%) of cells was positive for CD14 (monocyte and macrophage marker), CD19 (dendritic cell marker), CD34 (haemopoetic progenitor cell marker, and pan-leukocyte marker CD45. These results conform to previous reports for BMSCs cultured in HS [22].

### 2.2. Metabolic Activity and Cytotoxicity

BMSCs were plated at 5000 cells/well of 96-well plates (Corning) in 200 µL basic medium and allowed to adhere overnight. The next day, culture medium was changed to fresh basic medium with drugs added at the indicated concentrations. MM.1S myeloma cells were also plated at 5000 cells/well of 96-well plates in 200 µL RPMI-1640 (Sigma-Aldrich), and 10% fetal bovine serum (FBS, Gibco™) and 1% antibiotics with drugs added immediately at the indicated concentrations. Cells were cultured for 3 days before analysis of metabolic activity and cytotoxicity.

Metabolic activity of BMSCs and MM.1S cells was analyzed using the Cell Counting Kit-8 (CCK-8, Dojindo EU GmbH) according to the manufacturer’s protocol as previously described [21]. In brief, CCK-8 reagent was added to the culture medium (1:10) at the end of the culture period. The plate was incubated for 4 h at 37 °C, and the relative mitochondrial activity was measured in a Victor 1429 Multilabel Counter (Wallac; Turku, Finland) at 450 nm.

Cytotoxicity was analyzed by measuring the release of lactate dehydrogenase (LDH) from damaged cells. The analysis was performed from supernatant samples using Cytotoxicity LDH Assay Kit-WST (Dojindo EU GmbH, Munich, Germany) according to the manufacturer’s protocol.

### 2.3. Proliferation Assay

BMSC numbers in standard culturing conditions and adipogenic and osteogenic differentiation conditions were determined quantitatively by analyzing the total amount of DNA by CyQUANT Cell Proliferation Assay Kit (Thermo Fisher Scientific) as reported previously [23]. Briefly, BMSCs were lysed with 0.1% Triton-X 100 buffer (Sigma-Aldrich), and the supernatant was collected and stored at −80 °C until the final analysis. Fluorescence signals were measured with a Victor 1429 Multilabel Counter at 480/520 nm.

### 2.4. Indirect Co-cultures of BMSCs and Myeloma Cell Line MM.1S

For evaluating if the presence of BMSCs affects the cytotoxicity of melflufen in myeloma cells, BMSCs were indirectly co-cultured with myeloma cell line MM.1S. BMSCs have been previously reported to enhance the tumoricidity of doxorubicin and melphalan [4]. The tumoricidity of bortezomib, however, was reported to be reduced in the presence of BMSCs [24]. Therefore, bortezomib, as well as melphalan and doxorubicin, was used as a control drug in these experiments. BMSCs were plated on the bottom of the wells in 24-well plates at a density of 15,500 cells/cm^2^ (30,000 cells/well) in basic medium. Wells with only basic medium without BMSCs were used as controls. BMSCs were allowed to attach overnight before MM.1S cells were added to inserts (Thincert Cell Culture Inserts, PET membrane, pore diameter 0.4 µm) at 30,000 cells/insert in 100 µL RPMI-1640, 10% FBS, 1% antibiotics. After 72 h of co-culture, another 100 µL medium was added to the inserts with drugs making for a total of 200 µL medium in the inserts with drugs at the indicated concentrations. Cells were cultured with the drugs for another 72 h after which MM.1S cell viability was assessed using the CellTiter-Glo^®^ 2.0 Cell Viability Assay (Promega, Madison, WI, USA) according to the manufacturer’s instructions. Cell viability was measured in triplicate using a Victor 1429 Multilabel Counter.

### 2.5. Adipogenic Differentiation

BMSCs were placed at a density of 40,000 cells/cm^2^ in CellBind-treated 48-well plates (Corning Inc., Corning, NY, USA) in 500 µL basic medium. The next day, medium was changed to adipogenic differentiation medium consisting of DMEM/F12 (Thermo Fisher Scientific), 3% HS, 1% antibiotics, 1× Glutamax, 33 μM biotin (Sigma-Aldrich), 17 μM pantothenate (Sigma-Aldrich), 100 nM insulin (Sigma-Aldrich), 1 μM dexamethasone (Sigma-Aldrich), 0.5 mM isobutyl-methylxanthine (IBMX, Sigma-Aldrich), and 1 µM rosiglitazone (Sigma-Aldrich), and drugs were added at the indicated concentrations. The cells were induced for 7 days without medium change, after which medium was changed to maintenance medium, consisting of adipogenic differentiation medium without IBMX and rosiglitazone. Cultures were maintained for another 14 days with medium refreshed and drugs added twice per week. Samples were collected for analysis at the end of the culture.

Formation of lipid droplets was detected using Oil Red O as previously described [25]. In addition, glycerol 3-phosphate dehydrogenase (GPDH) activity was quantified as a measure of lipid metabolism using GPDH activity kit (Sigma-Aldrich) according to the manufacturer’s instructions. Absorbance at 450 nm proportional to the enzymatic activity was measured every 60 s for 1 h at 37 °C using a Varioskan Flash (ThermoScientific™). GPDH activity was normalized by total protein content determined using the Pierce™ BCA Protein Assay Kit (ThermoScientific™). Absorbance at 544 nm was measured with a Victor 1429 Multilabel Counter.

### 2.6. Osteogenic Differentiation

For osteogenesis, 250 BMSCs were placed in CellBind-treated 48-well plates (Corning Inc., Corning, NY, USA) in 500 µL basic medium. The next day, medium was changed to osteogenic medium consisting of basic medium supplemented with 200 μM ascorbic acid 2-phosphate (Sigma-Aldrich), 10 mM-glycerophosphate (Sigma-Aldrich) and 5 nM dexamethasone (Sigma-Aldrich). Drugs were added at the indicated concentrations. Cultures were maintained for 21 days with medium refreshed and drugs added twice per week. Samples were collected for analysis at the end of the culture. Osteogenesis was evaluated based on alkaline phosphatase (ALP) activity at 7 and 21 days as described previously [26]. To analyze mineralization, cultures were fixed at 21 days of culture with 70% ethanol and stained with Alizarin Red S (pH 4.1–4.3; Sigma-Aldrich) as previously described [26]. Mineralization was quantified by extracting the dye with 100 mM cetylpyridinium chloride (Sigma-Aldrich) and measuring the absorbance at 540 nm with a Victor 1429 Multilabel Counter.

### 2.7. Angiogenesis Assay

Vascular networks were established by co-culturing BMSCs with human umbilical vein endothelial cells (HUVECs) as described previously [27]. Briefly, BMSCs were suspended in endothelial growth medium-2 (EGM-2, Lonza) with 2% human serum (Serana Europe GmbH) and plated at 20,000 cells/cm^2^ in 48-well plates (ThermoFisher, Nunc™). After 2 h, green fluorescent protein (GFP)-labeled HUVECs (CellWorks) were added at 4000 cells/cm^2^ and cultured overnight in EGM-2. The medium was replaced by fresh EGM-2 the next day and supplemented with drugs at the indicated concentrations. Medium and drugs were refreshed at day 4, and samples were fixed in 4% paraformaldehyde after 7 days of co-culture.

### 2.8. Immunocytochemistry

Cultures were washed with PBS (Lonza), fixed and permeabilized for 15 min with 0.2% triton X-100 in 4% paraformaldehyde (Sigma-Aldrich), and blocked for 1 h with 1% bovine serum albumin (BSA; Sigma-Aldrich) in PBS. Osteogenically differentiated cells were stained overnight at 4 °C with primary antibodies for collagen I and osteocalcin, followed by incubation with secondary antibodies and 1 µg/mL TRITC-phalloidin (Sigma-Aldrich) for 45 min at 4 °C. Secondary antibody controls prepared in the absence of primary antibodies are presented in Appendix A.

Pericytes in the angiogenesis assay cultures were stained overnight at 4 °C with an antibody for α-smooth muscle actin (α-SMA) followed by incubation with a secondary antibody for 45 min at 4 °C. Controls for the used secondary antibody have been published before [28]. See Appendix A for the used antibodies and dilutions.

Counterstaining of nuclei with 0.5 µg/mL DAPI (Sigma-Aldrich) was performed at the end of the staining procedure in all samples.

Stained cells were imaged using an inverted microscope Olympus IX51 (Olympus, Tokyo, Japan) equipped with fluorescence unit and sCMOS camera (Orca Flash4.0LT, Hamamatsu, Hamamatsu City, Japan). Fluorescence images were taken using Alexa 488, Alexa 546, and DAPI filters, and 4 and 10x objectives. The image processing was performed with Fiji [29].

### 2.9. Quantification of Endothelial Cell and Pericyte Coverage Using Image Analysis

Parameters related to microvascular networks morphology and pericytic differentiation were quantified using Fiji software from the immunocytochemically stained angiogenesis assay cultures. For each condition, five donor cell lines were analyzed, with 3 culture wells per donor, and 2 regions of interest (1321 × 1321 µm^2^) from each well. Images were processed as described previously [28,30,31,32] with minor modifications. Briefly, brightness and contrast were adjusted by the “window/level” tool (35/70), and images were converted to a binary format by applying the “triangle” threshold method. The vascular area was quantified as the area positive for GFP, and the area covered by pericytes was defined as the area positive for α-SMA. The 2D “skeletonize” function was applied to calculate the total network length, and the average diameter was computed by dividing the total vasculature area by the total network length.

### 2.10. Quantified Real Time-Polymerase Chain Reaction (qRT-PCR)

Expression of osteogenic marker genes RUNX2a, DLX5 and SP7 and adipogenic marker genes LEP and FABP4 were quantified using qRT-PCR as previously described [33]. Briefly, total RNA was isolated from the cells at 21 days of differentiation with the Nucleospin kit (Macherey-Nagel GmbH & Co., KG, Düren, Germany), and RNA concentration was measured with a spectrophotometer (Nanodrop 2000, Thermo Fisher). First-strand cDNA was synthesized from the total RNA using the High-Capacity cDNA Reverse Transcriptase Kit (Applied Biosystems, Foster City, CA, USA). PCR reactions were performed in mixtures containing 50 ng cDNA, 300 nM forward and reverse primers, and SYBR Green PCR Master Mix (Applied Biosystems) on a QuantStudio 12K Flex Real-Time PCR System (Applied Biosystems) with initial enzyme activation at 95 °C for 10 min, followed by 45 cycles of denaturation at 95 °C for 15 s and anneal and extend at 60 °C for 60 s. Data were normalized to the expression of housekeeping gene RPLP0 (human acidic ribosomal phosphoprotein P0). The primer sequences (Oligomer Oy, Helsinki, Finland) and accession numbers are presented in Appendix A. Expression levels relative to BMSCs differentiated in the absence of chemotherapeutic drugs were calculated according to a previously described mathematical model [34].

### 2.11. Statistical Analyses

Statistical analyses were performed with R Statistical Software. Drug response curves were drawn and EC50 values estimated using the drc package [35] according to a three-parameter log-logistic function where the lower limit is equal to 0. The effect of drug concentrations on proliferation and indicators of differentiation were analyzed using Kruskal–Wallis one-way analysis of variance by ranks, followed by pairwise multiple comparisons with one control according to Dunn using the PMCMCRplus package [36]. The results were considered significant when the false discovery rate-controlled *p* value was below 0.05, represented as * *p* < 0.05, ** *p* < 0.01, *** *p* < 0.001.

## 3. Results

### 3.1. Viability, Cytotoxic Response, and Proliferation of BMSCs in the Presence of Standard-of-Care Drugs for Multiple Myeloma

We studied the effect of melflufen and common standard-of-care chemotherapeutic drugs melphalan and doxorubicin on BMSC viability using an assay for metabolic activity (CCK-8) and the cytotoxic response of BMSCs by measuring the release of LDH from damaged cells. BMSCs and MM.1S myeloma cells were cultured in standard culture conditions and exposed to drugs at various concentrations for 72 h, followed by analysis of viability (Figure 1a) and cytotoxicity (Figure 1b).

BMSCs were less sensitive to the tested drugs than MM.1S myeloma cells. The viability of MM.1S cells was consistently low at the tested drug concentrations except for the lowest concentration of doxorubicin (0.01 µM). BMSCs were more than 30 times as sensitive to melflufen (EC50 = 2.69 µM ± 0.06) than to melphalan (EC50 = 85.2 µM ± 1.2). The precise effect of melphalan on BMSC viability could not be determined: BMSC viability was also reduced by DMSO, which was used as vehicle control. The survival rate of cultures with melphalan was still lower than with the corresponding amounts of DMSO alone. The chemotherapeutic drugs were cytotoxic to the BMSCs in a dose-dependent manner. Culture with DMSO alone did not lead to the release of LDH by BMSCs. Because of the extremely low viability of the MM.1S myeloma cells, LDH release varied substantially between samples.

Proliferation of BMSCs cultured with the chemotherapeutic drugs was determined after 24 and 72 h by DNA-based CyQuant assay (Figure 2). At both time points, the EC50 drug concentration was about 18 times lower for melflufen than for melphalan. The EC50 values of both melflufen and melphalan dropped 3.4-fold between 24 and 72 h. Doxorubicin’s EC50 concentration dropped more than 10-fold between these time points, indicating that this drug has a slower mechanism of action compared to melflufen and melphalan.

### 3.2. Myeloma Cytotoxic Response in the Presence of BMSCs

To see if the presence of BMSCs affected the cytotoxic response of myeloma cells to melflufen and standard-of-care drugs, we established indirect co-cultures of MM.1S myeloma cells with BMSCs and evaluated MM.1S cell viability after 72 h of exposure to the drugs (Figure 3).

While co-culture with BMSCs protected MM.1S cells from doxorubicin and bortozomib-induced cell death, melflufen-induced MM.1S cell death was as efficient under co-culturing conditions as in MM1.S culture alone. The tumoricidal effect of melphalan, however, increased in the presence of BMSCs.

BMSCs did not affect the tumoricidal effect of melflufen. We continued by studying if melflufen could affect BMSC function, thereby modifying the microenvironment in which myeloma cells reside.

### 3.3. Adipogenic Differentiation

The proliferation rate of adipogenically differentiated BMSCs was affected by melflufen, melphalan and doxorubicin in a dose-dependent manner (Figure 4). Statistically significant reductions in BMSC number were found for all three drugs tested. BMSCs derived from one donor were not affected by melphalan in adipogenic differentiation conditions, illustrating the biological variance in drug response that can be observed.

We proceeded by evaluating the effect of the chemotherapeutic drugs on adipogenic differentiation marker expression using the same drug concentrations. Oil red O stainings showed the presence of lipid droplets in adipogenic differentiation conditions at 3 weeks of culture (Figure 5). The number of lipid droplets decreased with increasing concentrations of melflufen, melphalan and doxorubicin. DMSO alone seemed to lead to a decrease in the number of lipid droplets.

Glycerol-3-phosphate dehydrogenase (GPDH) is an enzyme involved in lipid biosynthesis. Its activity was reduced only at the highest concentrations of melflufen, melphalan and doxorubicin used in this study (Figure 6). This reduction in total enzymatic activity was found to be statistically significant.

Expression of adipogenic marker gene FABP4 at 3 weeks of culture was slightly decreased in the presence of melflufen and doxorubicin compared to control conditions (Figure 7). Adipogenic marker gene LEP was expressed at slightly lower levels when differentiated in the presence of melflufen and melphalan. The decrease in adipogenic marker gene expression levels was not found to be statistically significant.

### 3.4. Osteogenic Differentiation

In osteogenic differentiation conditions, culture with the drugs also led to a dose-dependent reduction in cell number (Figure 8). The BMSC population had halved after 3 weeks of culture with only 2 nM melflufen compared with the absence of drugs. The reduction in BMSC number was statistically significant only for 10 and 20 nM melflufen, however, due to substantial variability between donor cells lines as well as substantial intra-donor variability. For melphalan and doxorubicin, BMSC numbers showed a more gradual decrease with increasing drug concentration. Significant reductions in cell numbers were observed from 0.5 µM melphalan and 1 nM doxorubicin upward.

We continued analyzing the effect of the drugs on osteogenic differentiation using the same drug concentrations. After 3 weeks of culture in osteogenic differentiation conditions, mineralized matrix was visible in phase contrast images as dark speckles (Appendix A). The presence of the mineralized nodules decreased with increasing concentrations of melflufen, melphalan and doxorubicin. Apoptosis could be observed at the highest concentrations, as seen from a compromised morphology.

We analyzed activity of alkaline phosphatase (ALP), an enzyme involved in matrix mineralization. ALP activity decreased with increasing drug concentrations, with significant reductions observed at the highest drug concentrations tested (Figure 9). Statistically significant reductions of normalized ALP activity were found for 20 nM melflufen and 2 nM doxorubicin.

Mineralized matrix formation was assessed using Alizarin Red S stainings with osteogenically differentiated BMSC at 21 days of culture with drugs added at the indicated concentrations. A representative image of a culture well plate stained with Alizarin Red S stainings is presented in Appendix A. Quantification of the staining showed a dose-dependent decrease in mineralized matrix formation for all drugs (Figure 10), which is in line with our results for ALP activity. The amount of mineralized matrix was significantly reduced for 10 and 20 nM melflufen, 0.5 µM and 1 µM melphalan and 2 nM doxorubicin.

Immunostainings of the osteogenically differentiated BMSCs for collagen type I (Figure 11) and osteocalcin (Figure 12) confirmed the decrease in cell number with increasing drug concentrations. Collagen type I and osteocalcin was not distributed evenly in the culture wells, with some parts staining more strongly than others. For 0.5 µM melphalan and 1 nM doxorubicin, we could observe that part of the cells obtained an irregular shape and stained strongly for late differentiation marker osteocalcin, but not that strongly for collagen type I. Osteocalcin was abundantly expressed in the remaining cells at the highest drug concentrations.

In line with the immunostainings, gene expression levels of osteogenic marker genes RUNX2, DLX5 and SP7 were increased in the presence of melflufen, melphalan and doxorubicin (Figure 13). RUNX2 and DLX5 expression levels were significantly increased in the presence of 20 nM melflufen. For 1 µM melphalan, only RUNX2 expression was significantly upregulated. Doxorubicin at a concentration of 2 nM upregulated the expression of DLX5.

### 3.5. Angiogenesis

We also investigated the effects of the drugs on angiogenesis and pericytic activity of BMSCs. We established vascular networks through the co-culture of BMSCs with GFP-labeled HUVECs in angiogenic culturing conditions with drugs added after 1 day of culture (Figure 14). The cultures resulted in the formation of vessel-like structures by the HUVECs, supported by BMSC-derived pericytes marked by the α-SMA. Vessel structures clearly diminished with increasing concentrations for melflufen, melphalan and doxorubicin. In addition, the number of α-SMA-positive pericytes showed a dose-dependent decrease for all drugs tested, but to a lesser extent than HUVECs.

These observations were confirmed by our quantitative analyses, which showed a significant decrease in the area covered by GFP-HUVECs in the presence of all three chemotherapeutic drugs tested (Figure 15a). The α-SMA-positive area saw a dose-dependent decrease for all three drugs, but the decrease was significant only for 0.5 µM melflufen, the highest dose of melflufen tested (Figure 15b).

## 4. Discussion

BMSCs have an important role in the formation of the bone marrow microenvironments and are therefore essential in the pathogenesis of hematological diseases such as multiple myeloma [37]. Building evidence shows the importance of the microenvironment on MM therapeutic responses. It is therefore important to know if novel drugs such as melflufen target BMSCs.

Melflufen had an EC50 value for BMSC survival rate of 2.69 µM, and melphalan had an EC50 value of 85.2 µM. For melphalan, we observed cytotoxicity and loss in viability of BMSCs at concentrations above 50 µM. These concentrations are above the mean peak concentrations of melphalan observed in the plasma of patients undergoing high-dose chemotherapy treatments, which are between 15–40 µM [38].

The high potency of melflufen compared to melphalan is partially because of its high lipophilicity, which makes it diffuse rapidly through the cell membrane. Once inside the cell, melflufen is hydrolyzed by aminopeptidases and esterases to release the alkylating payload [16].

The higher cytotoxicity of melflufen compared to melphalan has been shown before in myeloma patient-derived plasma cells [39] and endothelial cells [40]. Melflufen was reported to have median EC50 values for survival of myeloma patient-derived plasma cells of 0.04 nM for newly diagnosed myeloma patients and 7.7 nM for relapsed myeloma patients [39]. Melphalan had survival EC50 values of 556 nM for plasma cells from newly diagnosed myeloma patients and 3193 nM for plasma cells from relapsed myeloma patients [39]. Both melflufen and melphalan are therefore more cytotoxic in myeloma patient-derived plasma cells compared to BMSCs. Melflufen’s novel mechanism of action is also more effective in myeloma cells than in BMSCs: we found that melflufen was 32 times more cytotoxic than melphalan in BMSCs, whereas the EC50 value for melflufen is 414 to 14,000 times lower than melphalan in myeloma patient-derived plasma cells.

Strese et al. reported IC50 values for the survival of bovine endothelial cells of 0.17 µM for melflufen and 42 µM for melphalan [40]. This means that melflufen was 247 times more effective than melphalan in bovine endothelial cells. This is a much bigger difference than the 32 times we found for BMSCs.

Efficacy of melflufen treatment in multiple myeloma is potentiated by various aminopeptidases and esterases, which hydrolyze melflufen to alkylating metabolites [39]. It is possible that BMSCs have different aminopeptidases and esterases expression and activity profiles, which could make them less sensitive to melflufen compared to myeloma cells and endothelial cells. Inhibition of aminopeptidases causes minimal inhibition of proliferation of BMSCs [41].

BMSCs are relatively resistant to many anti-cancer drugs [42,43,44], which was also seen with melphalan and doxorubicin [45]. Of the drugs tested, doxorubicin had the strongest anti-proliferative and apoptotic effects on BMSCs. Similar effects on BMSCs have been reported before [45,46]. Doxorubicin has been reported to activate the DNA damage response and cell cycle checkpoints, although BMSCs are relatively resistant to the cytotoxic effects compared to cancer cells [47].

For both melphalan and melflufen, the proliferation rate decreased at lower concentrations than the onset of cell damage and loss of viability: a significant loss of proliferation was observed at concentrations above 10 µM for melphalan and 0.1 µM of melflufen. These results are in accordance with Kemp et al., who observed that 50 µM melphalan did not cause any significant level of cell death to BMSCs after 48 h, but that the long-term expansion post-treatment was decreased [48]. In contrast, cyclophosphamide, another alkylating reagent, did not result in any reduction in MSC expansion post-treatment [49]. Effects similar to those for melphalan have been observed with cytostatum paclitaxel: BMSCs remained viable in the presence of paclitaxel but lost their ability to proliferate [43]. The authors suggested that blockage of the cell cycle at either G1 or G2 could reduce the apoptotic effects of paclitaxel, which had been previously shown for cancer cells [49]. A similar response to the DNA damaging effect of melphalan and melflufen could also save BMSCs from cell death.

We investigated the impact of BMSC co-culture on cytotoxic responses of the myeloma cell line MM.1S. BMSCs protected myeloma cells from bortezomib-induced cell death and increased melphalan-induced cell death, as has been reported previously [4,24]. BMSCs protected MM.1S cells from doxorubicin, contradicting previous finding in which BMSCs were shown to enhance the tumoricidity of doxorubicin [4]. The cytotoxic response of MM.1S myeloma cells to melflufen was not affected by the presence of BMSCs. However, the mechanism to why melflufen and melphalan differ in this respect is not known.

Many studies have demonstrated that MSCs support the drug resistance of myeloma cells: BMSCs protect myeloma cells from cell death induced by bortezomib [24] and dexamethasone [50]. Mechanisms through which BMSCs decrease the sensitivity of myeloma cells to drug treatments include the secretion of cytokines such as interleukin-6 (IL-6) and vascular endothelial growth factor (VEGF) [24,50], and cell-to-cell contacts [5]. Hao et al. showed that bortezomib-treated myeloma cells secrete miRNA-15a, resulting in cell cycle arrest and downregulation of VEGF expression [24]. BMSCs were shown to suppress miRNA-15a expression in myeloma cells, providing survival support and protecting myeloma cells from bortezomib-induced cell death.

BMSCs do not protect myeloma cells from all therapeutic agents. For melphalan and doxorubicin, BMSCs were shown to increase drug-induced cell death in myeloma cell line ANBL6 cells [4]. BMSCs significantly enhanced the tumoricidity of doxorubicin and melphalan by secreting IL-6, thereby suppressing bcl-XL expression and pushing ANBL6 cells into the cell cycle [4]. Proliferating myeloma cells likely become more sensitive to DNA-damaging agents in the presence of IL-6.

Our results are limited to a single myeloma cell line. Furthermore, because the cells were physically separated by the cell culture inserts, interactions between myeloma cells and BMSCs were limited to soluble factors. Direct cell-to-cell contacts between myeloma cells and BMSCs could possibly play a role in the therapeutic response to melflufen. BMSCs are known to offer greater protection from drug-induced apoptosis when they are in physical contact with the myeloma cells compared to when they are separated by a cell culture insert [4,5]. Cell cycle arrest protects myeloma cells from drug-induced apoptosis, and adherence to BMSCs may provide protection by inhibiting myeloma cell proliferation and accumulation in G_0_/G_1_ [5]. BMSCs separated from myeloma cells by a cell culture insert may however provide greater protection than the use of BMSC-conditioned medium, because nonactivated stromal cells may not produce factors protecting myeloma cells from drug-induced apoptosis [5].

Next, we investigated how the drugs affected differentiation of the BMSCs. Because adipose and bone tissues are present in the bone marrow, we decided to study how the drugs affect the formation of these tissues.

We observed a decrease in adipogenic differentiation in the presence of melflufen, melphalan and doxorubicin at the highest concentrations tested. The decrease in adipogenic markers was not only due to a decrease in cell number, as GPDH activity normalized to cell number was significantly reduced. Adipogenic differentiation of BMSCs has been reported to decrease in the presence of cytarabine, daunorubicin and vincristine, as seen from a reduction in Oil Red O staining [51], or after bleomycin treatment indicated by a reduction in BODIPY lipid staining [52]. The level of lipid stains in these reports was not normalized for cell number, however. Therefore, we cannot distinguish if the reduced lipid formation was due to an actual downregulation of adipogenic differentiation or due to a reduced cell number.

In osteogenic differentiation conditions, exposure to the drugs resulted in reduced formation of mineralized matrix. The reduction was in line with the decreased cell number, although ALP activity normalized to cell number also decreased with increasing drug concentrations. Somaiah et al. showed that cytarabine, daunorubicin and vincristine also decreased mineralized matrix formation in osteogenic differentiation conditions [51]. The results were not normalized for cell number, however; thus, the obtained results could be solely due to a reduction in cell number in the presence of the drugs. At the highest concentrations tested, we saw a marked upregulation of mature bone marker osteocalcin in the few cells remaining. This observation was confirmed by qRT-PCR, in which we saw upregulation of osteogenic marker genes at these concentrations.

We observed that for some donor cell lines, melphalan increased markers of adipogenic and osteogenic differentiation at low concentrations, including GPDH activity in adipogenesis and ALP activity and mineralized matrix formation in osteogenesis. The effect was only seen for some donor cell lines, which indicates that drug response is dependent on donor. We did not observe the same for melflufen, although both melphalan and melflufen are alkylators. We cannot exclude that increased differentiation markers would be observed at even lower concentrations of melflufen than the ones tested, or by adjusting the frequency at which the drug is added [53].

In growing tumors, the nutrient and oxygen deficiency triggers an “angiogenic switch” to induce normal vasculature to sprout [54]. The increase in vascular formation in tumors contributes to tumor cell proliferation, growth, and metastasis [55]. Anti-angiogenic drugs can benefit cancer patients, as they can reverse the angiogenic switch and disturb the tumor’s blood supply. In this work, we studied the effect of melflufen on in vitro angiogenesis. In vitro angiogenesis assays were formed through the co-culture of HUVECs with BMSCs. The BMSCs act as pericytes that support angiogenesis [28].

Melflufen, melphalan and doxorubicin caused a dose-dependent decrease in the areas covered by endothelial cells and pericytes. In line with our results, a previous study on the cytotoxicity of melflufen in endothelial cells showed a dose-dependent inhibition of vascular tube formation [40]. In addition to a cytotoxic effect in monocultures of bovine endothelial cells and HUVECs, the authors also showed that both melflufen and melphalan inhibited tube formation in co-cultures of HUVECs with fibroblasts [40]. The effect of the drugs on the pericytic activity of the fibroblasts was not assessed separately. Melphalan is known to have potent anti-angiogenic effects in HUVEC and endothelial progenitor cells [56]. Melphalan may trigger endothelial vascular toxicity by deregulation of Myc and NF-kb1 transcription factor intracellular pathways [57]. Exposure to melphalan impairs the capacity of endothelial cells to form vascular structures on Matrigel, an effect that is highly dependent on the treatment regime [56].

We found quite strong anti-angiogenic effects of doxorubicin compared to what has been previously reported [58,59,60]. Uvez et al. found a significant decrease in HUVEC viability at only 0.2 µM doxorubicin [59]. In the current study, we found a significant reduction in the area covered by HUVECs at a 10 times lower concentration. However, the experimental set-ups differ not only in analysis method and presence of BMSCs in our set-up, but also in duration: HUVECs were cultured in the presence of the drugs for 48 h by Uvez et al. [59], where our co-cultures were exposed to drugs for 6 days. Doxorubicin did not have a strong inhibiting effect on tube formation of HUVECs on Matrigel up to 1 µM [60]. However, again, these results are difficult to compare to the ones obtained in our experimental setting due to the difference in time scale. HUVECs only take 16 h to form tubes on Matrigel [60], which is a fast process compared to our angiogenesis assay, that takes one week.

The cytotoxic effect was more pronounced in endothelial cells than in pericytes, especially for melflufen. BMSCs are less sensitive to melflufen and melphalan than what has been reported for HUVECs [40]. In addition, melflufen’s mechanism of action is more effective in HUVECs than in BMSCs because of the expression of aminopeptidase-N [40], in which the expression is upregulated in endothelial cells within mouse and human tumors [61]. Aminopeptidase-N plays important roles in angiogenesis by supporting capillary tube formation, cellular motility and adhesion [62].

## 5. Conclusions

Here, we studied the cytotoxicity of melflufen in healthy human BMSCs and how melflufen affects BMSC proliferation and differentiation into adipocytes and osteoblasts. Due to its novel mechanism of action, BMSCs are more sensitive to melflufen than to melphalan. The cytotoxicity of melflufen in myeloma cells was not affected by the co-culture with BMSCs, as was the case for melphalan, bortezomib and doxorubicin.

Adipogenesis, osteogenesis and BMSC-mediated angiogenesis were all affected by melflufen. Melphalan and doxorubicin affected BMSC differentiation in similar ways as melflufen. The effects of the drugs on adipogenesis and osteogenesis were not solely because of effects on proliferation, seen from the differential expression of differentiation markers normalized by cell number.

These results indicate that melflufen has a significant effect on BMSC proliferation and differentiation. Possible effects on BMSCs should therefore be assessed when evaluating treatment outcome.

## Figures and Tables

**Figure 1 cells-11-01574-f001:**
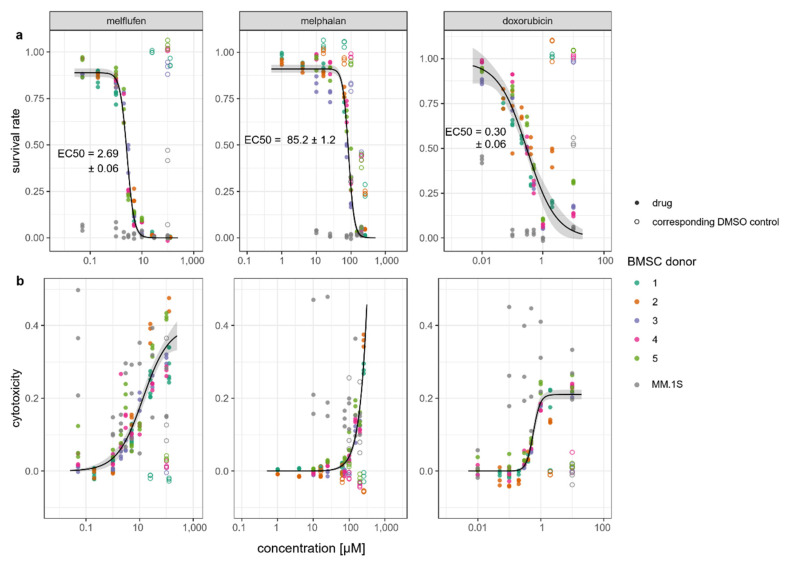
Viability of bone marrow-derived stem/stromal cells (BMSCs) and MM.1S myeloma cells in the presence of drugs. Survival rates (**a**) and cytotoxic response (**b**) of BMSCs and MM.1S myeloma cells after 72 h of culture with melflufen, melphalan and doxorubicin at indicated concentrations relative to control cultures without drugs. Dots represent individual culture wells with a color-coded donor cell line. Open circles indicate dimethylsulfoxide (DMSO) vehicle controls for the indicated drug concentrations. Doseؘ–response curves for BMSCs with confidence interval indicated in gray.

**Figure 2 cells-11-01574-f002:**
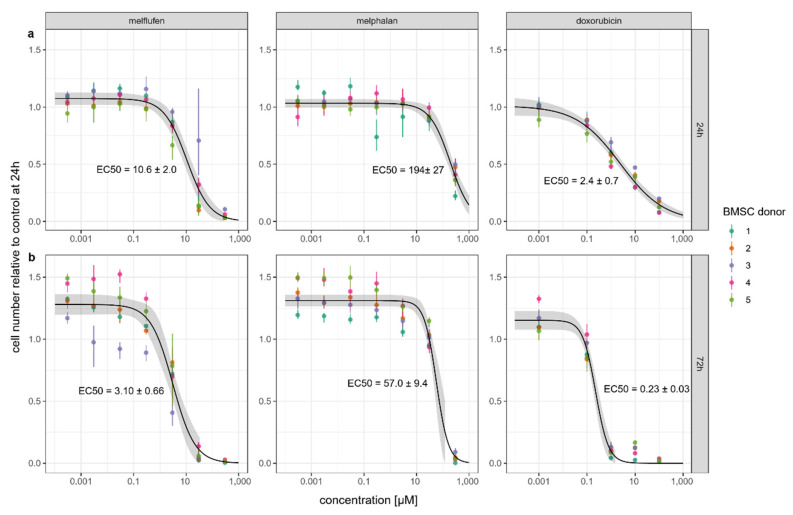
BMSC proliferation in the presence of drugs. Relative number of BMSCs cells at 24 h (**a**) and 72 h (**b**) of culture with drugs at indicated concentrations. Cell number expressed as relative to the control without drugs at 24 h. Dots represent the mean of a color-coded donor cell line cultured in triplicates with crossbars indicating the measured range. Dose–response curve with confidence interval indicated in gray.

**Figure 3 cells-11-01574-f003:**
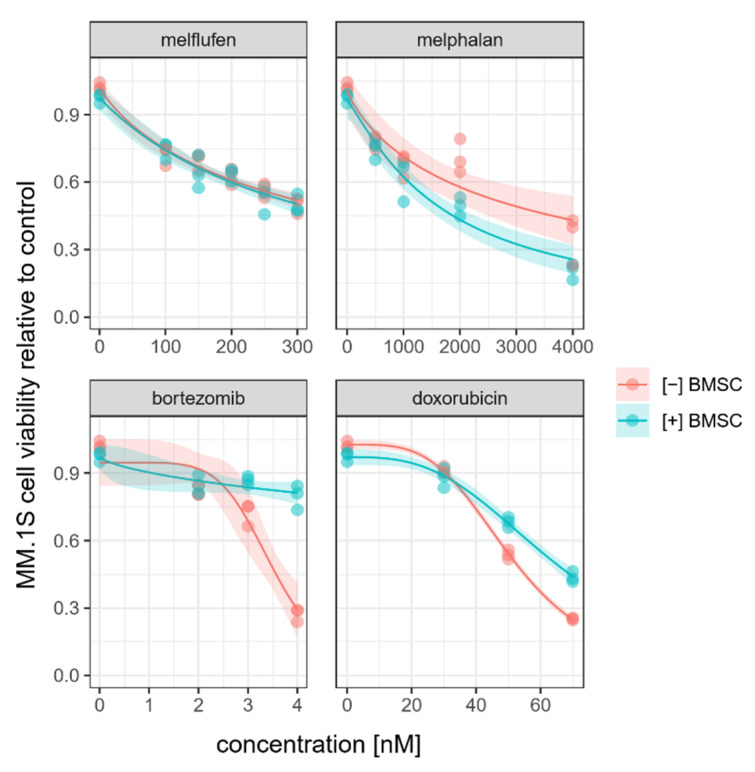
Effect of BMSCs on cytotoxic response of myeloma cells to drugs. Relative viability of MM.1S myeloma cells after 72 h of culture in the presence of melflufen, melphalan, bortezomib, and doxorubicin in the presence (blue) or absence (red) of BMSCs in indirect co-cultures. Results are shown from a single experiment out of the three experiments run in total with different BMSC donor cell lines. Dots represent individual culture wells. Dose–response curves are flanked by confidence intervals indicated in gray.

**Figure 4 cells-11-01574-f004:**
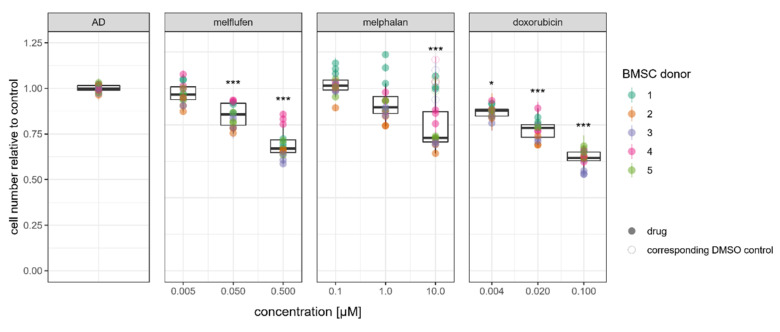
Drug effect on BMSC proliferation in adipogenic differentiation conditions. Relative numbers of BMSCs at 21 days of culture in adipogenic differentiation conditions with melflufen, melphalan, and doxorubicin added at indicated concentrations. Dots represent individual culture wells with a color-coded donor cell line and crossbars indicating the measured range. Open circles indicate DMSO vehicle controls for 10µM melphalan. * denotes *p* < 0.05 and *** denotes *p* < 0.001 with Dunn’s test.

**Figure 5 cells-11-01574-f005:**
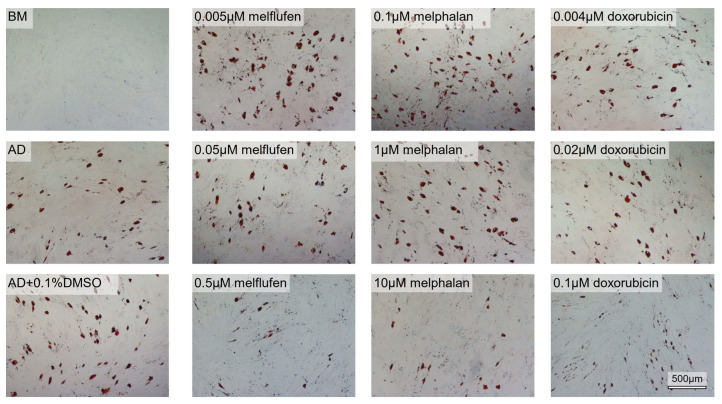
Fat droplet formation in adipogenic differentiation of BMSCs in the presence of drugs. Oil Red O stains lipid droplets red in BMSCs after 3 weeks of culture in adipogenic differentiation (AD) conditions with melflufen, melphalan, doxorubicin, and DMSO added at indicated concentrations. BM, basic medium control. Scale bar: 500 µm.

**Figure 6 cells-11-01574-f006:**
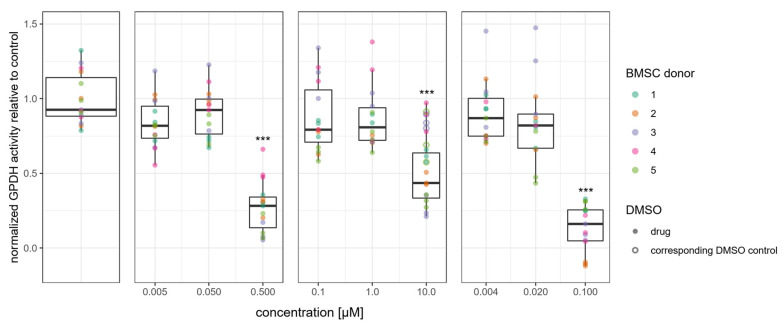
Drug effect on lipid biosynthesis in adipogenically differentiated BMSCs. Activity of glycerol-3-phosphate dehydrogenase (GPDH) at 21 days of culture in adipogenic differentiation conditions with melflufen, melphalan and doxorubicin added at indicated concentrations. GPDH activity is normalized by total protein content and is presented relative to the control. Dots represent individual culture wells with a color-coded donor cell line. Open circles indicate DMSO vehicle controls for 10 mM melphalan *** denotes *p* < 0.001 with Dunn’s test.

**Figure 7 cells-11-01574-f007:**
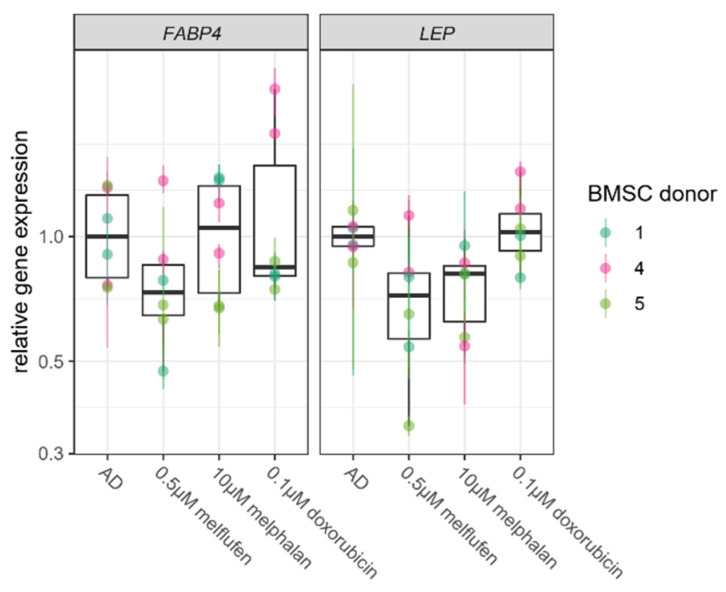
Drug effect on expression of adipogenic marker genes. Relative expression levels of adipogenic marker genes FABP4 and LEP in BMSCs at 21 days of culture in adipogenic differentiation conditions with 0.5 µM melflufen, 10 µM melphalan and 0.1 µM doxorubicin. Gene expression levels are shown relative to the mean expression in adipogenically differentiated control without drugs (AD). Dots represent individual culture wells with a color-coded donor cell line, and crossbars indicate the measured range.

**Figure 8 cells-11-01574-f008:**
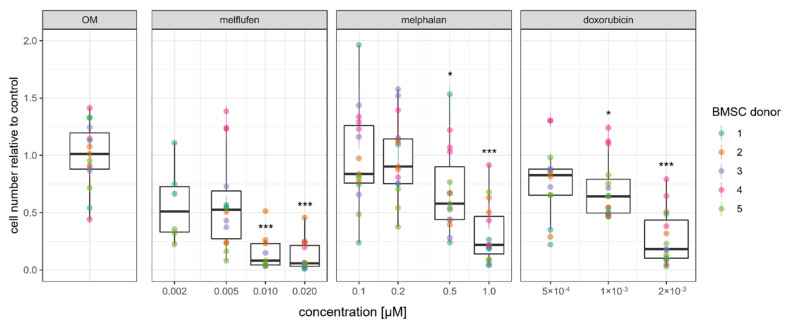
Drug effect on BMSC proliferation in osteogenic differentiation conditions. Relative numbers of BMSCs at 21 days of culture in osteogenic differentiation conditions with melflufen, melphalan, and doxorubicin added at indicated concentrations. Dots represent individual culture wells with a color-coded donor cell line. * denotes *p* < 0.05 and *** denotes *p* < 0.001 with Dunn’s test.

**Figure 9 cells-11-01574-f009:**
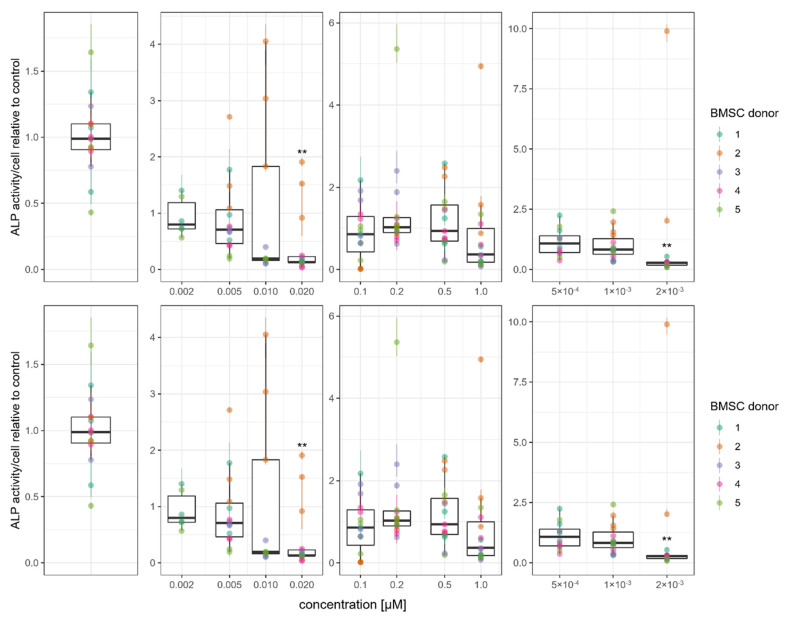
Drug effect on bone formation in osteogenically differentiated BMSCs. Alkaline phosphatase (ALP) activity of BMSCs at 21 days of culture in osteogenic differentiation conditions with melflufen, melphalan, and doxorubicin added at indicated concentrations. ALP activity is normalized by total cell number and is presented relative to the control. Dots represent individual culture wells with a color-coded donor cell line. OM, osteogenic medium control without drugs. ** denotes *p* < 0.01 with Dunn’s test.

**Figure 10 cells-11-01574-f010:**
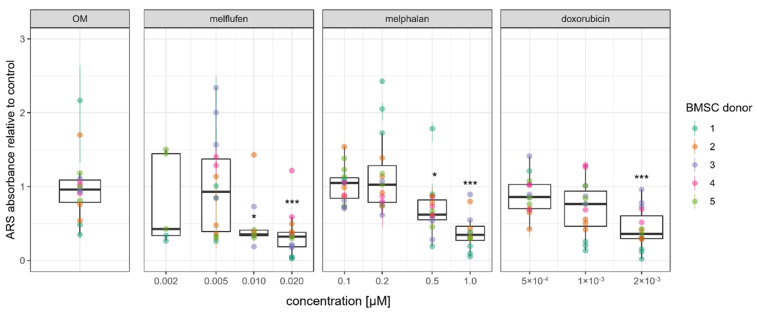
Mineralized matrix formation by osteogenically differentiated BMSCs in the presence of drugs. Quantification of the Alizarin Red S (ARS) stainings presented relative to the control. Dots represent individual culture wells with a color-coded donor cell line. * denotes *p* < 0.05 and *** denotes *p* < 0.001 with Dunn’s test.

**Figure 11 cells-11-01574-f011:**
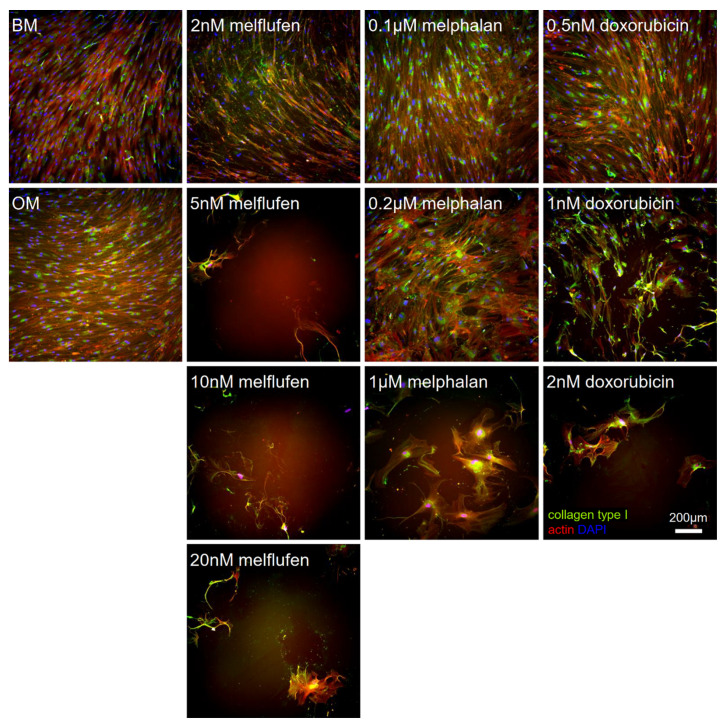
Drug effect on the expression of collagen type I in osteogenically differentiated BMSCs. Immunofluorescent staining for collagen type I (green) in osteogenically differentiated BMSCs at 21 days of culture with melflufen, melphalan, and doxorubicin added at indicated concentrations. Counterstained with phalloidin for actin filaments (red) and DAPI for nuclei (blue). BM, basic medium control; OM, osteogenic medium control without drugs. Scale bar: 200 µm.

**Figure 12 cells-11-01574-f012:**
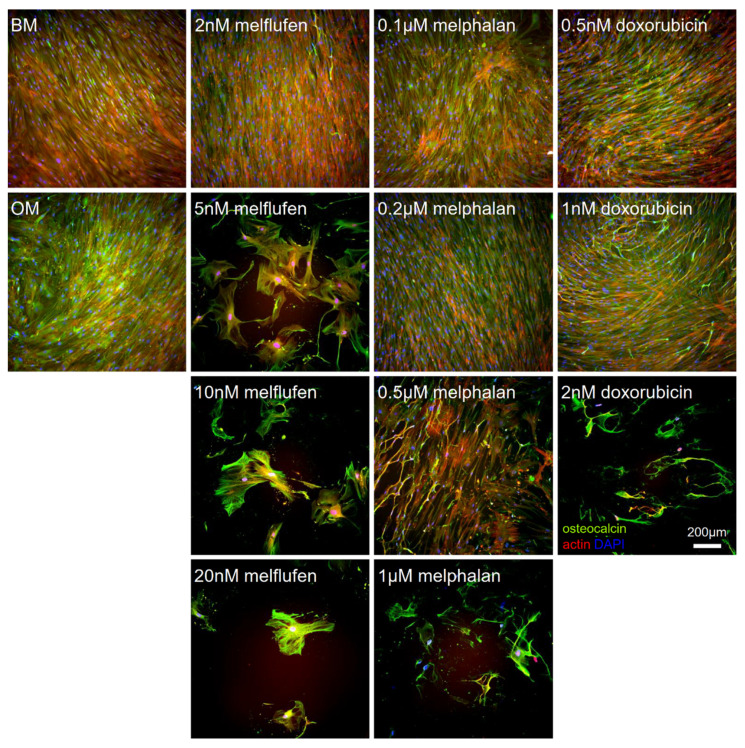
Drug effect on expression of osteocalcin in osteogenically differentiated BMSCs. Immunofluorescent staining for osteocalcin (green) in osteogenically differentiated BMSCs at 21 days of culture with melflufen, melphalan, and doxorubicin added at indicated concentrations. Counterstained with phalloidin for actin filaments (red) and DAPI for nuclei (blue). BM, basic medium control; OM, osteogenic medium control without drugs. Scale bar: 200 µm.

**Figure 13 cells-11-01574-f013:**
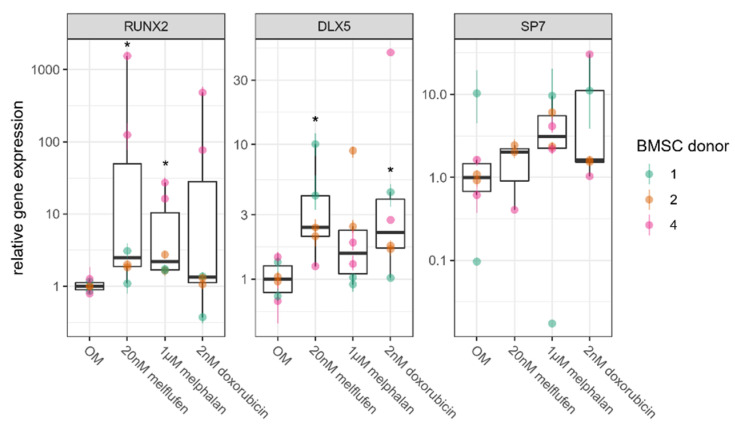
Drug effect on the expression of osteogenic marker genes. Relative expression levels of osteogenic marker genes RUNX2, DLX5 and SP7 in BMSCs at 21 days of culture in osteogenic differentiation conditions with 20 nM melflufen, 1 µM melphalan and 2 nM doxorubicin. Gene expression levels are shown relative to the mean expression in osteogenically differentiated control without drugs (OM). Dots represent individual culture wells with a color-coded donor cell line and crossbars indicating the measured range. * denotes *p* < 0.05 with Dunn’s test.

**Figure 14 cells-11-01574-f014:**
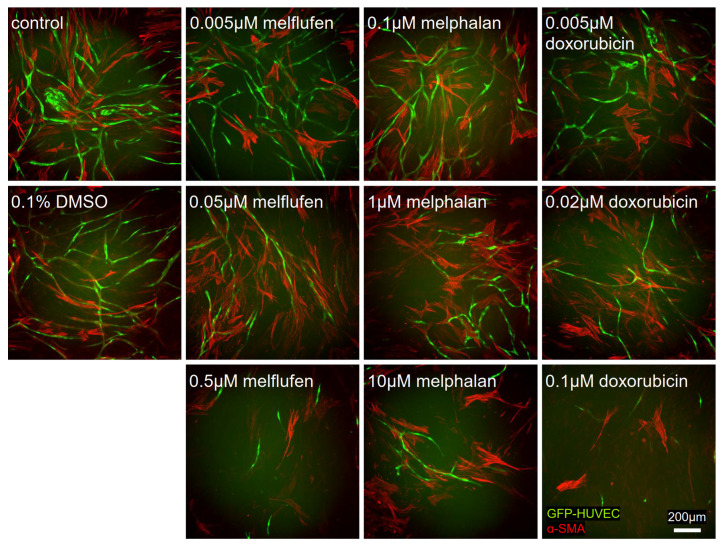
Drug effect on vascular network formation. Vascular networks formed by co-culturing green fluorescent protein-labeled human umbilical vein endothelial cells (GFP-HUVECs, green) and BMSCs for 7 days in the presence of drugs or DMSO at the indicated concentrations. BMSCs are stained for pericytic marker α smooth muscle actin (α-SMA) shown in red. Scale bar: 200 µm.

**Figure 15 cells-11-01574-f015:**
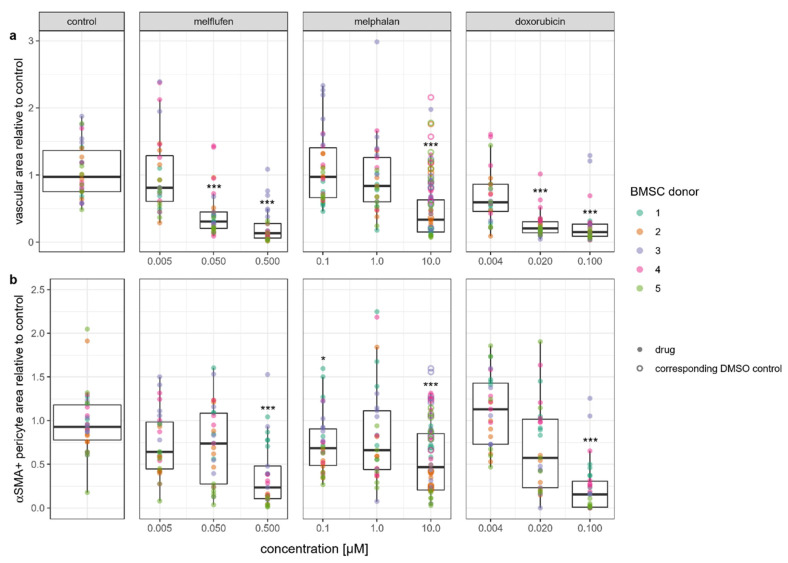
Quantitative analysis of vascular network formation in the presence of drugs: (**a**) quantification of vascular area covered by GFP-HUVECs; (**b**) pericyte area covered by α smooth muscle actin (α-SMA)-positive BMSCs. Dots represent regions of interest from co-cultures involving color-coded donor BMSC lines with melflufen, melphalan, and doxorubicin added at indicated concentrations. Open circles indicate DMSO vehicle controls for 10 µM melphalan. * denotes *p* < 0.05, *** denotes *p* < 0.001 with Dunn’s test.

## Data Availability

Data available on request.

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
