# Peer review of "Growth Response and Differentiation of Bone Marrow-Derived Mesenchymal Stem/Stromal Cells in the Presence of Novel Multiple Myeloma Drug Melflufen"

_cells, 2022, doi:10.3390/cells11091574_

Round 1

Reviewer 1 Report

The manuscript entitled "Growth response and differentiation of bone marrow-derived 2 mesenchymal stem/stromal cells in presence of novel multiple 3 myeloma drug melflufen" by Gebraad and colleagues presents a very comprehensive study on the effect of melflufen treatment on BMSCs.

Several points need to be addressed before publication:

-line 97: Apoptois specific assay is not reported

-drugs screening should be removed from the manuscript since this part does not provide any advantage to the study

-Please provide a more detailed description of the drugs tested (melphalan and melflufen)

-Line 234: bortezomib treatment is reported only in the final experiment of the study. This should be explained

-Lines 257-258: please delete this sentence

-Paragraph 3.1 the assays used to evaluate viability, apoptosis and proliferation of treated BMSCs should be indicated

-Lines 281-282-. This sentence is not clear, please rephrase

-Line 285 The terms "BMSCs numbers" is not clear. Did you mean proliferation by CyQUANT assay?

-Paragraph 3.1 Authors should justify the differences in drug concentrations used in the adipogenic differentiation assay. Maybe figure 3 is not clear and should be modified

-Figure 7 Non normalized data may be misleading

-Figure 12 these data should be moved at the beginning of the paragraph

-Line 496 This sentence "BMSCs were more sensitive to melflufen than to melphalan" is redundant.

Reviewer 2 Report

The paper by Gebraad et al. describes the responses of bone marrow-derived mesenchymal stem/stromal cells (BMSC) from healthy donors to melflufen, a new drug for the therapy of multiple myeloma (MM). The authors compare the effect of melflufen (a form of melphalan modified so as to be more active, due to its high lipophilicity), to melphalan and doxorubicin (plus controls such as DMSO). The action of melflufen is also tested on one MM cell line, MM.1S. Different types of action of melflufen are examined, all very carefully, using different approaches : cell viability assays ; proliferation assays ; adipogenic and osteogenic differentiation assays ; angiogenesis assays ;  immunochemistry studies ;  and gene expression assays (including qRT-PCRs). Melflufen is also studied in co-cultures of BMSC and MM.1S cells. Hence the studies of the effect of melflufen on BMSC and MM.1S are quite complete, rather exhaustive. Figures are easy to read. The results are clear, though limited (and expected) : BMSC are significantly more sensitive to melflufen than to melphalan (Nb : doxurubicin has the strongest toxic effect on BMSC). In addition, BMSC do not protect MM.1S from the toxic effects of melflufen (while BMSC protect MM.1S cells from doxorubicin-induced cell death). The discussion of these results is well conducted, including the potential negative effects of the strong inhibition of BMSC by melflufen, in addition to the inhibition of myeloma plasma cells. In conclusion, the study was performed appropriately and provides useful information, thus deserves to be published in Cells.

lines 257-259 should be deleted.

Author Response

To reviewer 2:

Thank you very much for your comments. The use of a single myeloma cell line in the BMSC-myeloma cell co-cultures is a limitation, which we have now briefly discussed.  Lines 257-259 have been deleted.

Yours sincerely,

The authors

Reviewer 3 Report

The study demonstrated the important point of interest in the field of drug resistance in MM patients. The methodology and the manuscript were well writen. The result is also of interest for further study the efficacy in using melflufen in combination with other drugs classes to enhance the response and overcome drug resistance producing by BM stromal cell. 

Author Response

Thank you very much for your comments. 

Yours sincerely,
The authors

Reviewer 4 Report

Since the interest of the study is to investigate if BMSCs increase or decrease the tumnoricidial effect of melflufen, I suggest the Authors to firstly report the data regarding the cocultures of BMSCs with MM.1S myeloma cells and then those regarding the activity of different drugs on adipogenesis, osteogenesis and BMSC-mediated angiogenesis which could explain the possible mechanisms of action of melflulen in comparison with other chemotherapeutic drugs. Furthermore, they should more extensively discuss the limitations of their model of BMSC coculture with the myeloma cell line MM.1S.

Author Response

Thank you very much for your comments.

We have moved the data regarding the co-culture of BMSCs and the myeloma cell line before those regarding adipogenesis, osteogenesis and BMSC-mediated angiogenesis.

We have extended the discussion on the limitations of our model of BMSC coculture with a single myeloma cell line, which were physically separated by the cell culture inserts.

Yours sincerely,

The authors

Round 2

Reviewer 1 Report

Authors addressed all the points raised by the reviewer. The manuscript is now acceptable for publication.

Reviewer 4 Report

In the present form the aim of the study is better specified and the data are more clearly and precisely reported.